# Network Modeling Approaches and Applications to Unravelling Non-Alcoholic Fatty Liver Disease

**DOI:** 10.3390/genes10120966

**Published:** 2019-11-24

**Authors:** Montgomery Blencowe, Tilan Karunanayake, Julian Wier, Neil Hsu, Xia Yang

**Affiliations:** 1Department of Integrative Biology and Physiology, University of California, Los Angeles, 610 Charles E. Young Drive East, Los Angeles, CA 90095, USA; montyblencowe@g.ucla.edu (M.B.); tmkaru@g.ucla.edu (T.K.); julianrwier@g.ucla.edu (J.W.); neilhsu2000@gmail.com (N.H.); 2Molecular, Cellular, and Integrative Physiology Interdepartmental Program, University of California, Los Angeles, 610 Charles E. Young Drive East, Los Angeles, CA 90095, USA; 3Interdepartmental Program of Bioinformatics, University of California, Los Angeles, 610 Charles E. Young Drive East, Los Angeles, CA 90095, USA

**Keywords:** network modeling, integrative genomics, NAFLD, NASH, steatosis, systems biology, multi-omics

## Abstract

Non-alcoholic fatty liver disease (NAFLD) is a progressive condition of the liver encompassing a range of pathologies including steatosis, non-alcoholic steatohepatitis (NASH), cirrhosis, and hepatocellular carcinoma. Research into this disease is imperative due to its rapid growth in prevalence, economic burden, and current lack of FDA approved therapies. NAFLD involves a highly complex etiology that calls for multi-tissue multi-omics network approaches to uncover the pathogenic genes and processes, diagnostic biomarkers, and potential therapeutic strategies. In this review, we first present a basic overview of disease pathogenesis, risk factors, and remaining knowledge gaps, followed by discussions of the need and concepts of multi-tissue multi-omics approaches, various network methodologies and application examples in NAFLD research. We highlight the findings that have been uncovered thus far including novel biomarkers, genes, and biological pathways involved in different stages of NAFLD, molecular connections between NAFLD and its comorbidities, mechanisms underpinning sex differences, and druggable targets. Lastly, we outline the future directions of implementing network approaches to further improve our understanding of NAFLD in order to guide diagnosis and therapeutics.

## 1. Introduction

The liver is a central organ involved in vital biological processes including synthesis of key products such as cholesterol, bile acids, triglycerides and glycogen, immune processes, detoxification, and serum homeostasis. Pathological changes within these key liver pathways are a common healthcare issue, accounting for around 2 million deaths per year globally [1]. One specific subset, non-alcoholic fatty liver disease (NAFLD) is becoming an increasingly common issue affecting around 25% of the world’s population on average [2]. This disease encompasses an array of pathological conditions starting from benign steatosis (i.e., fat accumulation in the liver) to non-alcoholic steatohepatitis (NASH) which features inflammation and cell damage, to mild scarring of the liver (fibrosis), widespread scarring and distortion (cirrhosis), and hepatocellular carcinoma (HCC) (Figure 1). Currently there are no FDA-approved therapeutic options, partly due to the limited knowledge we have of the mechanisms involved in the progression of the disease, as well as the differing environmental and genetic contributions that each individual may possess [3]. As a result, there has been an ever-increasing demand for liver transplantation, which is now the second most common form of organ transplantation, with NASH predicted to overtake Hepatitis C as the main cause of transplantation between 2020 and 2030 [4,5].

To understand NAFLD pathogenesis, one must explore the multidimensional complexities of the disease where clinical differences and similarities have been spotted in disease manifestations between individuals, and explore the intricate molecular details involved in disease progression. First of all, NAFLD has distinct genetic and environmental determinants. The genetic heritability estimates of NAFLD range from about 20% to 70% [6]. Recent Genome-wide Association Studies (GWAS) have identified several genetic loci that are significantly associated with NAFLD susceptibility. These loci include *PNPLA3*, *TM6SF2*, *MBOAT7,* and *GCKR*, each of which have been replicated across studies with ethnic, geographic, and methodological variations [7,8]. In terms of environmental contributions, previous studies have heavily indicated obesity and its role in accelerating the epigenetic age of the liver [9], as well as sleep deprivation [10], diet composition [11], maternal high fat intake [12], and type 2 diabetes as risk factors [13]. These risk factors likely interact with genetic susceptibility loci to culminate in disease incidence. It is currently unclear what contributes to the interindividual differences in NAFLD progression and overall outcomes in patients, as only around 40% of them progress to liver fibrosis, and of those, 20% continue to develop advanced fibrosis and cirrhosis [14].

Secondly, although the liver is the focal organ which manifests the disease burden, one must not disregard the crosstalk between additional organs in the development of NAFLD. Current understanding suggests that gene mutations in the hypothalamus, particularly leptin signaling, can contribute to NAFLD and obesity [15]; gut permeability differences can result in the leak of pro-inflammatory signals that contribute to NAFLD [16,17]; resistance to insulin in adipose tissue can result in the redistribution of fat to other organs, including the liver [18]; and the release of adipokines and cytokines among others contributing to NAFLD [19]. Given the connections of the adipose tissue, hypothalamus, and the gut with the liver, we can delve deeper into each organ’s potential contribution to further dissect key tissue-tissue interactions and the underlying molecular mechanisms [15,20].

The third complexity of NAFLD is that numerous comorbidities exist, such as metabolic syndrome (MetS), type 2 diabetes (T2D), obesity and cardiovascular pathologies. Moreover, there are also documented changes in brain physiology as evidenced by micro and macro cerebrovascular alterations, and an increased likelihood of stroke, lesions and cognitive impairment [21]. Given these connections, we can explore the overlaps in pathogenic pathways across diseases as well as the “chicken and egg” idea to discover if one disease is causal to the other and identify shared causal drivers and networks in disease progression [22]. Additionally, NAFLD exhibits sexual dimorphism, with a greater NAFLD predisposition in men and post-menopausal women [23]. In animal models of NAFLD, a higher degree of proinflammatory/profibrotic cytokines are identified in males [24]. The mechanistic underpinnings of sexual dimorphism remain underexplored.

Given the complexities discussed above, NAFLD is a multifactorial disease that poses challenges within the context of the path towards biological understanding and therapy. Uncovering the perturbation of genes and their molecular counterparts within particular tissues, which lead to a pathological disruption of biological homeostasis, is progressively becoming a key focus of scientific research. Traditional approaches that examine one gene or factor at a time have become less efficient for addressing the multidimensional complexities of NAFLD.

With the advancement of high throughput omics technologies and analytical pipelines, network modeling has emerged as a powerful tool to help integrate multidimensional information to elucidate the complex molecular systems within and between tissues in the pathogenesis of NAFLD and other associated diseases. Moreover, the rise of high throughput single-cell sequencing technologies is enabling a better dissection of distinct cell populations to reveal more elusive molecular regulatory interactions within and between cell types. Both tissue and cell level molecular networks have the capacity to better capture the molecular underpinnings of disease. This is particularly relevant with the increasing recognition of the omnigenic disease model, which predicts that numerous genes that are highly connected in molecular networks underlie the functions of each biological system and when dysregulated, will lead to complex diseases such as NAFLD [25]. 

In this review, we will first present our current knowledge of NAFLD via non-network approaches and the remaining gaps from these more traditional methods. We will then briefly cover the principles behind the common network modeling methods and focus on summarizing their applications to understanding NAFLD through the integration of multi-omics datasets as well as their potential to delve into possible preventative and therapeutic strategies. Lastly, we will outline the current limitations and future directions of the network applications in NAFLD research.

## 2. Current Understanding of NAFLD and Remaining Gaps

The progression of NAFLD is highly heterogeneous, and therefore, solving the mechanisms behind the development from mild to severe fatty liver disease is not a straightforward process (Figure 1). Many pathogenic drivers of NAFLD are unlikely to be the same across patients but it is thought that the liver at some point is overwhelmed with lipid build up which subsequently causes hepatocellular stress and results in fibrogenesis [26,27,28]. Specifically, free fatty acids (FFAs) are thought to be key to the pathogenesis of NASH, whereby they are delivered to the liver via the blood from triglyceride lipolysis and de novo lipogenesis from excess carbohydrates [29,30]. FFAs are commonly reduced by two processes: re-esterification into triglycerides and mitochondrial beta-oxidation [31]. However, when these processes are perturbed and FFAs accumulate, they form lipotoxic lipids, causing inflammasome activation, oxidative stress, and ER stress leading to apoptosis and cytokine/chemokine release, and eventually leading to fibrosis via hepatic stellate cell activation and increasing Kupffer cell density [32,33,34]. Some of the most common contributing factors to FFA overload is insulin resistance within adipose tissue (common in T2D), excess dietary sugars, and to a lesser extent cholesterol, uric acid, the gut microbiome, and sleep apnea [35,36,37,38]. 

Recent genetic studies of NAFLD have offered clues to the potential causal genes and pathways involved. The heritability estimates of NAFLD generally range from 20% to 70% depending on the ethnicity of those tested and overall study design [6]. When considering the indices of disease severity, the heritability estimate measured via twin studies was 0.52 for hepatic steatosis based on MRI proton-density fat fraction and 0.5 for liver fibrosis based on liver stiffness [39]. To more clearly define individual genetic risks of diseases, GWAS is a powerful tool that has been implemented over the years to find associations between genetic regions (loci) and certain traits/diseases, in this case NAFLD. Several genetic variants have been identified via GWAS to have a strong effect size and thus are likely to significantly contribute to NAFLD heritability. These loci include *PNPLA3*, *TM6SF2*, *MBOAT7,* and *GCKR* [7,8]. These candidate genes are involved in lipid homeostasis and metabolic pathways. Of these, associations have been found particularly with *PNPLA3* (patatin-like phospholipase domain–containing 3) [40,41] and *TM6SF2* (transmembrane 6 superfamily member 2) [42]. *PNPLA3* encodes for a protein that is involved in glycerolipid hydrolysis and lipase activity, as well as the metabolism of monoacylglycerol, diacylglycerol, and triglycerides. There are implications that changes to the morphology of this protein may lead to the inability of substrate binding, resulting in the disruption of triglyceride hydrolysis, lipid droplet accumulation, and thus liver disease. *TM6SF2* has been shown to regulate hepatic triglyceride and serum low density lipoprotein levels, while also being a marker for fibrosis/cirrhosis [42,43]. Moreover, *in vivo* studies report that knocking down *TM6SF2* in the livers of mice resulted in a threefold increase in hepatic steatosis and a reduction in the circulating levels of cholesterol and plasma triglycerides [42]. More recently a splice variant in the gene *HSD17B13* that encodes the hepatic lipid droplet protein has shown an association with reduced Alanine Transferase (ALT) and Aspartate Aminotransferase (AST), implying less inflammation/injury due to producing a nonfunctional protein. Overall, the significant GWAS loci uncovered to date only explain a small fraction (<5%) of the total genetic heritability for NAFLD, implying that numerous additional loci and genes with moderate to subtle effects as well as epistasis and interactions with environmental factors are yet to be discovered. Thus, major advances can still be made within the context of this disease at the genetic level.

Studies of the interactions between genetic risk factors and environmental risk factors such as high fat and atherogenic diets in mouse models also revealed genes and pathways underlying diet-induced liver steatosis and NASH. Using inbred mouse strains, Hui et al. first investigated hepatic triglyceride accumulation in ~100 genetically divergent mouse strains fed a high fat and high sugar diet, observing large variations in liver steatosis across the mouse strains of different genetic background [44]. Via a GWAS, they identified three genetic regions with potential causality and highlighted *GDE1* as a previously unknown mediator of triglyceride homeostasis. Expressional modulation of this gene confirmed this locus as a regulator of hepatic triglyceride accumulation, with overexpression increasing triglyceride content and knockdown having an opposite effect. *GDE1* encodes glycerophosphodiester phosphodiesterase 1, an enzyme involved in the degradation of glycerophosphoethanolamine and glycerophophocholine. One of the products of *GDE1* is glycerol-3-phosphate, a substrate for triglyceride biosynthesis. *GDE1* overexpression led to an increase in *Cd36* expression in the liver, which can enhance lipid transport. These observations suggest that *GDE1* may regulate triglyceride accumulation by increasing substrate availability and triglyceride flux. In another study, the same group examined western diet fed mice carrying hemizygous transgenes for human apolipoprotein E*3-Leiden and cholesteryl ester transfer protein to model human NASH progression [45]. Again, they noted significant differences among mouse strains in the transition from simple steatosis to NASH, and via a GWAS they highlighted several genetic loci and candidate genes including *PHACTR2*, involved in inflammation and cell cycle control, and *FUCA1* which regulates cell growth and signaling. Moreover, they identified several pathways shared between mice and human NASH subjects, including innate and adaptive immune system, cell cycle, Notch signaling, TGFβ signaling, and Wnt signaling. Together, these two studies highlight the complex interactions between environmental factors (diet in this case) and causal genetic loci underlying different stages of NAFLD.

Despite some basic understanding of the cellular and molecular mechanisms in NAFLD progression and potential targets of NAFLD, the following questions have not been thoroughly addressed: What are the overall genetically, as opposed to environmentally driven mechanisms for the development and progression of NAFLD? Which tissues, genes, and processes differentiate the different stages of NAFLD? Are there sex-specific pathogenic processes? Are these mechanisms tissue-specific and are there inter-tissue interactions? What explains the interconnections between NAFLD and other cardiometabolic diseases? Given the lack of sensitivity of imaging or existing biomarkers, what are the biomolecular predictors of NAFLD? Addressing these knowledge gaps will significantly improve our ability to design new diagnostic, preventative, and therapeutic strategies against NAFLD and its associated complications. 

## 3. Importance of Omics Data in Offering Integrated Network Views of Complex Diseases 

To provide comprehensive answers to the above questions, traditional approaches that examine one factor at a time have become less efficient. With the ever-increasing omics technologies available, we now have the essential tools to unravel the hidden complexities of this multifactorial disease (Figure 2). Omics datasets range from genomics, epigenomics, transcriptomics, proteomics, metabolomics and metagenomics [46,47,48]. 

Genomics, the study of the genome (the DNA sequence), enables us to understand the structures, regulation, organization, and functions of different genes. It also can provide insights into how genetic variants can contribute to disease or differing levels of function of a particular process. Epigenomics examines the non-genetic contributions towards cellular function, traits, or phenotypes. These can occur through processes such as DNA methylation, histone modification, chromatin organization, and non-coding RNAs (such as microRNAs (miRNAs) or long non-coding RNAs (lncRNAs)), which can downregulate or upregulate gene expression and functions. Transcriptomics, the study of the transcriptome, measures mRNA levels to understand gene expression changes that occur in cells or tissues in response to genetic or environmental stress. This component is the culmination of the genetic and epigenetic landscape which determines whether a specific locus is capable of undergoing transcription. While this data may give some indication of the key genes involved in the pathogenesis, transcriptome data represents changes in gene expression levels, and thus does not necessarily portray protein and metabolic fluxes within a tissue accurately.

At one level above transcriptomics, proteomics aims to measure protein expression levels as well as various types of post-translational modifications that determine tissue functions. The proteome is particularly important as proteins execute biological functions and are a major class of pharmaceutical targeting and can thus be advantageous in drug development. Whereas metabolomics investigates the low molecular weight compounds that are substrates, intermediates, or products of metabolic reactions in our biological system, such as inorganic ions, organic acids, amino acids, sugars, nucleotide and nucleosides, vitamins, carbohydrates, peptides, aromatic amines, and lipids. Many proteins are key enzymes catalyzing metabolic reactions, and metabolites can in turn participate in diverse molecular and cellular functions ranging from epigenetic regulation of gene expression to protein modification to metabolic reactions. Finally, metagenomics, the genome of microbial communities, is another important area of investigation as the link between the microbiome and health is increasingly evident. The microbiome can form symbiotic relationships with the host, participating in nutrient digestion, absorption, immune response, and even neuronal regulation. 

As indicated above, each omics layer only reflects a specific aspect of the molecular cascades and most studies that examine a single omics layer will only reveal fragmented views. Thus, consideration of the interplays between these different components constituting our physiological systems is critical. Only by integrating data across omics domains (multi-omics integration), can a holistic understanding of how a disease may develop be derived (Figure 2). 

One particularly powerful approach to integrate multi-omics data to better understand the pathophysiology of NAFLD and to pinpoint pharmacological targets for treatment is network modeling (Figure 2) [49]. Networks form a natural framework to elucidate the relationships between the large number of molecules measured by multi-omics methodologies. In a network, biological components such as genes, proteins, or phenotypes are depicted as nodes and the relationship between these nodes as edges. The edges confer information regarding gene regulatory interactions, enzymatic reactions, or statistical relationships between the nodular components. In essence, the collection of nodes and edges graphically display a gene network and can be used to express both networks within a particular tissue/cell type or between two or more tissues/cell types of interest. Therefore, networks informed by different factors, such as genetic versus environmental risks, males versus females, steatosis versus NASH, and NAFLD versus other diseases, can be compared to address fundamental questions governing this disease and the complex molecular mechanisms modulating the initiation and progression of pathology. More importantly, networks exhibit graphical topologies and most biological networks are arguably considered to possess a scale-free topology, meaning that the fraction of nodes with degree (number of connections) *k* follows a power law *k*^-^^α^. In a scale-free topology, a few important nodes, termed “hubs”, have many connections, or a high degree of connectivity, whereas most other non-hub nodes have few connections and form the peripheral nodes around the central node. Network nodes that bridge high numbers of shortest path connections between other nodes are considered “bottleneck hubs”. Such topological features thus can potentially help prioritize novel druggable targets. In particular, hub genes are likely more pathologically relevant as they have a higher degree of connectivity, and thus likely have a larger effect on phenotypes.

## 4. Commonly Used Network Models

There are numerous types of molecular networks that capture different types of molecular interactions. The commonly used ones include gene regulatory networks (GRNs) that focus on regulatory relations among genes, protein-protein interaction (PPI) networks that depict physical interactions among proteins, metabolic networks that illustrate metabolic cascades, and hybrid networks that combine different types of interactions.

### 4.1. Tissue and Single Cell GRNs

GRNs generated using bulk tissue analyses can be classified into the following methods as detailed in a previous review: regression, correlation, mutual information, ordinary differential equations, Bayesian approaches and Gaussian graphical models [49]. We will focus on the methods found in NAFLD studies. Correlation-based methodologies such as WGCNA [50] and MEGENA [51] are gene co-expression networks used to identify clusters of highly coregulated genes, termed “modules”, that can then be used to highlight the pathways and predict hub genes for therapeutic targeting [50,51,52]. Of the two, WGCNA has been more commonly implemented but the more recent MEGENA has the benefit of assigning a gene to more than one cluster and defining smaller and more coherent clusters rather than large sized modules built with WGCNA. Gene co-expression networks are straightforward, efficient, and valuable tools to broadly partition the transcriptome into subsets of genes that are coregulated and involved in similar functions. However, co-expression does not directly indicate directional regulatory relations between genes within or between modules, thereby limiting the dissection of detailed gene-gene regulation. Gaussian graphical models (GGMs) uses partial correlation of one gene in relation to another gene conditioning on other network members to estimate undirected, linear relationships between two genes to represent gene-gene interactions [53,54]. GGMs are computationally efficient and can infer potential interactions between genes but lack directional information. Finally, Bayesian Network (BN) approaches use the conditional dependency values of genes in respect to its parent nodes to create directed acyclic graphs (DAGs), which effectively visualizes regulatory relationships between genes [49]. The main limitations of BNs include higher computation cost and that optimal topology may not be detected. Therefore, each network method comes with unique advantages and disadvantages. There have been various iterations or adjustments to these network principles catering for the needs of particular labs, to help answer certain questions or improve the speed of analysis [55].

With the heterogeneous nature of tissues, single cell sequencing technologies have increased over the years to capture how cells interact to perform higher level functions within the tissue. As summarized in Blencowe et al., there has been a recognized need for new GRN modeling approaches required for single cell data [56], due to the lack of success in the common bulk tissue based network models [57]. Gene networks within a cell population or between cell-cell communication networks can be built based on various algorithms or assumptions. The most commonly used approaches to infer cell-cell interactions build on known ligand-receptor pairs, assuming cells expressing a ligand will interact with another cell expressing the corresponding receptor. Single cell network modeling is in its infancy in general, and improved methods are still greatly needed to deal with the novel biological questions asked by single-cell data.

### 4.2. Protein-Protein Interaction (PPI) Networks

PPI networks are defined as physical contacts between two or more protein molecules. They are often modeled through graphs where nodes represent proteins and edges represent undirected and potentially weighted physical contact. These contacts can be binary, which measure interaction between pairs of proteins or co-complex, in which both the direct and indirect interactions between groups of proteins are measured. Reliability information linked with corresponding interactions can be incorporated through edge weights. These networks can be constructed from a variety of sources, the majority of which are either manually constructed from known interactions (e.g., MINT [58], HPRD [59], BioGRID [60]), computationally constructed associations, or a combination of the two (STRING [61]) which combine known and predicted interactions to maximize the quality and quantity of coverage [58,59,60,61,62].

### 4.3. Metabolic Networks

Genome Scale Metabolic Models (GEMs or GSMMs) describe stoichiometry based metabolic reactions through the construction of gene-protein-reaction associations via both experimental and genome annotations data [63]. These associations may include assigning localization, energy estimation, identifying linked reactions and determining stoichiometry. These models enable metabolic flux predictions for an entire series of metabolic cascades via flux balance analysis and the use of linear programming. Another important feature of GEMs is that they can integrate various omics data types with kinetics/PPI networks, thus generating a more comprehensive biological picture. To date comprehensive networks have been built for specific areas of research and can continue to improve with increased data. Generic human GEMs include the Human Metabolic Reaction database (HMR) [64], HMR2 [65], and Recon2 [66]. Importantly, a number of liver specific GEMs have been constructed including HepatoNet1 [67] and iHepatocytes2322 [65], which are useful for liver focused research such as NAFLD.

### 4.4. Literature-based Pathways and Networks

Ingenuity Pathway Analysis (IPA), a commercial tool, takes advantage of the rising availability of high throughput gene-expression data and prior biological knowledge to create comprehensive molecular networks [68]. Rather than conventional network models that only infer interactions, IPA utilizes algorithms based on Ingenuity Knowledge Base (IKB), an immense collection of experimental data, curated literature, and third-party databases to create nodes and linking edges that denote directionality. Through the use of a series of stepwise algorithms applying an asymptotic Gaussian approximation of Z scores, statistically significant upstream regulators to downstream inducers of biological function/disease can be inferred [68].

### 4.5. Hybrid Networks

STITCH (Search Tool for Interactions of CHemical) is a unified database which employs Pearson correlation to understand the interactions of small molecules (i.e., drugs/chemicals) and corresponding proteins at the molecular/cellular level, thereby gaining information on metabolic pathways and drug efficacy [69]. Subsequently, protein-chemical interactions are made by combining chemical-chemical interactions with protein interactions (such as STRING networks) and text-mining imported protein-chemical interactions (PDSP *K*i Database and Protein Data Bank). Moreover, additional proteins/metabolites can be extrapolated by intersecting resultant protein-chemical interactions with annotated pathway databases (KEGG, Reactome, etc.). Metacore is another commercial integrative tool that allows the functional analysis of multiple omics data (metabolomics, proteomics, transcriptomics, and genomics) to generate network models [70]. Metacore’s own curated knowledge database is also more extensive (1,662 vs. IPA’s 662) and selective as it only consists of human protein-protein, protein-DNA interactions, transcription factors, signaling and metabolic pathways, disease and toxicity, and the effects of bioactive molecules [70,71]. Another hybrid model, Minimal Network Enrichment Analysis (MiNEA), uses transcriptomic, proteomic, and metabolomic data to isolate deregulated minimal networks corresponding to metabolic groups and associated metabolites, thus enabling the comparison of two experimental conditions. By constructing metabolic tasks which contain a set of disease-associated metabolites and utilizing thermodynamic constraints, the MiNEA algorithm can be used to identify pathogenic variations in metabolites [72]. Finally, Mirwalk aims to capture miRNA-target interactions and provide a framework for pairing these regulatory interactions with pathways, diseases, and gene/protein data. Additionally, while miRNAs regulate gene expression by affecting the intergenic regions, they also interact with lncRNAs, thus this framework also includes targets and non-targets for these interactions [73]. 

### 4.6. How to Determine What Type of Networks to Use 

As discussed, these various types of network models can reveal different molecular interactions and hence serve as complementary methodologies and resources for addressing the various biological questions of NAFLD (Figure 2). In general, what type of network to use depends on several factors, including the overall goal of the study, the data types at hand, and the computational capacity. For instance, if genetic and transcriptomic data and gene-gene interactions are the main focus, GRNs will be the most appropriate. For proteomics data, PPI networks are necessary. When metabolomics data are included, GEMs should be considered. Literature-based and hybrid networks are applicable in many cases, but they are usually very dense and may lack specificity or relevance to a particular study. For instance, when tissue specific interactions are investigated, use of non-tissue specific hybrid network models may add irrelevant interactions from other tissues. Ideally, multiple types of networks should be considered for a given study. For example, to explore gene-gene interactions, we have found that two complementary types of GRNs are particularly informative and deliver experimentally validated gene-gene relations. The first category is gene co-expression networks such as WGCNA [50,74] and MEGENA [51]. The second category is Bayesian networks that can flexibly incorporate various types of prior information such as genetic causality, genetic regulation of gene expression (expression quantitative trait loci or eQTLs), transcription factor binding, and epigenetic regulatory information to model directional but sparse regulatory networks [75,76,77,78,79,80]. Combination of these two types of networks is particularly powerful in revealing novel mechanistic insights, as one offers a global view of gene co-regulation and the other provides sparse but directional granular regulatory relations to elucidate many different contributors to disease pathogenesis [75,76,77,81,82,83,84,85,86,87,88,89,90,91,92,93,94,95,96].

## 5. Use of Network models to understand the pathogenic mechanisms in NAFLD development and progression

### 5.1. Modeling of single tissue networks in NAFLD

Several network methods have been applied to resolve the pathogenic alterations in molecular processes and biological pathways involved in the various stages of NAFLD progression (Table 1). These network applications have identified several key pathways that are perturbed during NAFLD progression, particularly including changes in extracellular matrix (ECM) structure, metabolic processes, and immune signaling (Figure 3).

#### 5.1.1. ECM Structure in Liver Tissue

Through the use of WGCNA, a co-expression network modeling approach, to identify top modules and genes in NAFLD via transcriptomics, Lou et al. found several top hub genes including *LUM*, *THBS2*, *FBN1,* and *EFEMP1*, all of which were significantly upregulated in advanced fibrosing-NAFLD across several human cohorts and in ApoE−/− mice [103]. Upregulated *LUM* expression is associated with hepatic fibrosis involving collagen fibrillogenic and increased ECM turnover [120]. Although not directly linked to advanced NAFLD, under expressed *EFEMP1* has been implicated in HCC [121], and both *FBN1* and *THBS2* are constituents in cellular adhesion and ECM that are essential for healthy liver physiology [122,123]. Interestingly, ECM and PI3K were the top pathways identified in the fibrosis driver module and PI3K signaling is known to regulate ECM deposition, collagen synthesis and the expression of pro-fibrogenic factors, all of which can drive disease progression via ECM remodeling in a feed-back manner [103].

In another study focusing on the cause of liver fibrosis, Zhan et al. utilized gene expression data from the Gene Expression Omnibus (GEO [124]) from patients with hepatitis B/C and NAFLD to identify differentially expressed genes (DEGs) [125]. In order to analyze the connections from the proteins encoded by the DEGs, the STRING database was used to construct the PPI network. Here, they identified 25 hub genes that were shared across hepatitis and NAFLD that involve fibrogenesis, many of which have been previously suggested including *LUM*, *FBN1, THBS2* in the study discussed above [103], *COL1A1,* and *COL6A3*, which were implicated in a previous miRNA-gene regulatory network analysis [109]. Utilizing the LINC1000 drug repositioning tool they further predicted that Zosuquidar compound and its target gene *ABCB1*, may have antifibrotic activity [125]. A similar analysis by Qi et al [107]. utilizing the HPRD-derived PPI liver network construction of mild and severe NAFLD patients also identified pathways and hub genes critical in ECM structuring and tissue connectivity, particularly ubiquitin 4 (*UBQLN4*), which regulates epithelial cell formation and survival [107]. Moreover, the finding of sex-hormone binding globulin (*SHBG*), a player in male-sex hormone deficiency, as a candidate gene is interesting as it reaffirms NAFLD overlap with endocrine processes as well as suggesting a sex-specificity that is discussed later in the review [107]. 

#### 5.1.2. Metabolic Pathways in Liver and Adipose Tissues

Metabolic disruption is a significant factor in NAFLD, with a wide breadth of individual pathways identified through network modeling. To better understand the molecular pathophysiology of NAFLD with respect to FFAs and lipids in general, Sahini et al. examined the differences in gene expression involved in lipid droplet formation of patients with bland steatosis (without hepatocyte ballooning, inflammation or fibrosis) versus hepatosteatosis and modeled these DEGs via a STRING PPI network construction. Of the 146 lipid droplet-linked DEGs, 51 were associated with liver-receptor homolog-1 (NR5A2), a key regulator in cholesterol homeostasis, bile synthesis, and triglyceride metabolism [106]. Particularly regarding fatty acids, an *in vitro* hepatocyte culture comparison of NASH versus bland steatosis hepatocytes revealed DEGs associated with mitochondrial carnitine palmitoyltransferase 1A (CPT1A), the enzyme that initiates fatty acid oxidation [126]. Moreover, these steatotic primary human hepatocyte cultures demonstrated induction of lipid droplet-associated *PLIN2*, *CIDEC*, *DNAAF1* and suppressed expression of *CPT1A*, *ANGPTL4*, and *PKLR* (Table 1) all of which are implicated in mitochondrial metabolism and functionality [106]. 

Another study by Pandey et al. recently implemented a multi-omics (transcriptome, proteome, and metabolome) based Minimum Network Enrichment Analysis (MiNEA) on mouse and human liver samples [72]. Their aim was to find all minimal subnetworks in liver from a given metabolic process and then by using deregulated genes between two conditions, the analysis can provide minimal deregulated networks. They found similarities between mouse and human networks for NASH key regulators in ceramide and hydrogen peroxide synthesis, as well as differences in the deregulation networks of NASH in phosphatidylserine synthesis [72]. This tool may be useful in exploring the many different metabolic phenotypes categorized under NAFLD to help elucidate species-specific processes for the best translational value.

Additionally, several liver specific GEMs have been constructed to facilitate the study of the pathophysiology of hepatic disease within the context of metabolic variation [67,127]. One of the more prominent findings via the use of these liver specific GEMs comes from a series of studies by Mardinoglu et al. in which hepatocyte specific data from the Human Metabolic Reaction 2.0 database and proteomics data (Human Protein Atlas) were used to construct iHepatocytes2322 [65]. Using this computational system, they created personalized GEMs based on metabolic data from patients with varying degrees of hepatic steatosis revealing several key metabolic loci which are perturbed in these patients. In particular, they report decreased serine and glycine levels as a result of NAD+ and glutathione insufficiency, which was confirmed further with additional evidence [65,101]. First, decreased glutathione/glutathione disulfide ratios were indicated in NAFLD patients [128]. Secondly, serine supplementation improved hepatic steatosis and reduced plasma levels of markers for liver damage (ALT, AST, and alkaline phosphatase) after supplementation [101,129].

To achieve a more comprehensive understanding of inter-tissue dynamics at play in NAFLD progression, Shubham et al. studied Visceral Adipose Tissue (VAT) [100]. Here, they implemented the WGCNA method to organize the disease transcriptome data from the VAT of NAFLD patients into co-expression modules of different processes and pathways. To delve further into the metabolic processes behind the disease, they built a network of metabolic genes in the Human Metabolic Network HMR2 and then compared this to the co-expression network built based from the varying genes. Additionally, they used previous information from adipocyte metabolism as a form of verification for reporter metabolites. Through this analysis, it was found that there was overlap between metabolic processes and inflammatory activity in NAFLD patients with sphingolipid and arachidonic acid metabolism genes being co-expressed in inflammatory pathways. Interestingly, they suggest potential fibrosis biomarkers of NASH being sphingosine, ceramide and their metabolites due to the time-dependent gene changes arising in NASH with fibrosis patients [100]. The findings on ceramides agree with previous evidence that adipose tissue ceramides are increased in equally obese individuals with higher liver fat content [130,131]. 

#### 5.1.3. Immune System in Liver and Adipose Tissues

Several network modeling studies have implicated immune dysregulation as a potential causal factor in NAFLD. In the study by Haas et al. both immune cell profiling and WGCNA were used to provide empirical evidence for cellular changes and capture hepatic gene sets associated with NASH in obese patients with certified NASH [112]. Their computational analysis showed that coregulated gene groups involved in antigen presentation, inflammation, T cell activation and cytotoxic responses are involved in NASH progression [112]. Similar results were seen in Sahini et al. in which PPI (STRING) and transcription factor networks coupled with DEG analysis were used to identify the importance of TLR4 signaling, B-lymphocyte chemokine and activator, *CXCL13* and *STAT4*, and the lipid phosphohydrolase *PPAP2B*, thus further linking transcriptional dysregulation of immune signaling and liver disease [106]. 

STRING PPI networks were also used to identify key markers of specific immune cells, where hub genes upregulated in the liver of NAFLD mice included *CD68* and *CTSS* in macrophages and leukocyte markers *PTPRC* and *ITGAX*, suggesting that cell adhesion processes play an important role in NAFLD within the context of the pathogenic immune response [102]. Additionally, other network models have uncovered loci associated with immune pathway perturbation particularly in the VAT as a large amount of fat that accumulates here has the potential to migrate to the liver via the portal vein. Shubham et al. utilized transcriptome-based Bayesian modeling to find strong inflammatory links including *PTGS2* and *ALOX5*, which indirectly contribute to inflammatory chemotaxis and further the progression from steatosis to NASH [100,132,133]. Clinical studies have previously shown that inhibitors of *PTGS2* and *ALOX5* could be an effective treatment in NAFLD [134]. Focusing on the later stage of disease, Chan et al. aimed to determine the differences between a cirrhotic liver and a healthy liver through DEG microarray analysis and utilized a hybrid network model (MetaCore) to determine the function and pathway of the DEGs [108]. They found unique gene expression patterns related to cell growth, inflammation, and the immune response, including 18 upregulated genes such as *ITGA2*, *ELF3* and *OAS2*, and one down regulated gene, *IL1RAP*, which is important in initiating the activation of interleukin 1 responsive genes [108].

#### 5.1.4. Coordination of ECM, Metabolism, and Immune Pathways in Liver

Coupling bulk tissue and cutting-edge single cell transcriptomic studies, Xiong et al. revealed the coordinated activities of the pathways/processes discussed above [105]. They found that the expression of genes involved in lipid metabolism and oxidative reactions was suppressed following diet-induced NASH in mice, while NASH-induced genes were highly enriched for the pathways responsible for ECM remodeling (*Col1a1*, *Mmp12*), cell adhesion, phagocytosis, and immune response (*Ccr2*, *H2-ab1*, and *Lcn2*). Single cell RNA sequencing of non-parenchymal cells in NASH mice revealed NASH-associated macrophages (NAMs) which showed elevated *Trem2*, encoding triggering receptors expressed on myeloid cells 2 and linked with both mouse and human NASH with increased disease severity [105]. To date we only note one single cell analysis for NAFLD, and this landmark study opens a new research direction which will enable a higher resolution understanding of the role of individual cell types in the heterogeneous progression of the disease.

### 5.2. Network Modeling of Multiple Tissues

To understand NAFLD at a more macroscopic scale, another method using network approaches can come using multi-omics datasets to investigate the molecular cascades behind the potential multi-tissue contributions towards disease. To understand the causal mechanisms underlying NAFLD, Krishnan et al. carried out a multi-tissue multi-omics integrative study using the Hybrid Mouse Diversity Panel (HMDP [135]), consisting of ~100 strains of mice with various degrees of steatosis [82]. By integrating GWAS, tissue-specific transcriptome, multiple types of network modeling approaches (WGCNA, MEGENA, Bayesian networks), and corresponding tissue-specific expression quantitative trait loci (eQTLs), they identified pathogenic processes that are liver specific (peroxisome, oxidative phosphorylation, Notch signaling), adipose specific (innate immunity, insulin signaling) or involving both liver and adipose (adaptive immune system, diverse lipid metabolism processes, apoptosis/cell cycle). Using the Bayesian network topology, they then predicted candidate regulatory genes of these NAFLD processes, including *THRSP*, *PKLR*, and *CHCHD6* in the liver, and *FASN* in both adipose and liver tissue. *In vivo* knockdown experiments of the candidate regulatory genes in liver improved both steatosis and insulin resistance. Further *ex vivo* testing demonstrated that downregulation of two novel regulators predicted, *PKLR* and *CHCHD6*, lowered mitochondrial respiration and led to a shift toward glycolytic metabolism in the liver mitochondria, thus highlighting mitochondria dysfunction as a key mechanistic driver of NAFLD in the liver [82]. Interestingly, human NAFLD GWAS genes were found to be peripheral genes rather than hub genes in these tissue-specific gene networks, suggesting that common genetic variations contributing to human NAFLD are not essential network genes but more likely to be disease modifiers as implicated in the omnigenic disease model.

Similarly, in a study by Lee et al. where they integrated PPI and transcriptome data to model co-expression networks in human liver, adipose, and muscle tissues that are associated with NAFLD, *PKLR* was identified as a potential target gene involved in liver fat accumulation [110]. They additionally found that *PNPLA3* (a top human GWAS hit) and *PCSK9* were important in contributing to liver steatosis and more acutely to HCC and can serve as druggable targets for NAFLD given the disease’s wide spectrum (Figure 1) [110]. Interestingly, *PCSK9* is a druggable target for cholesterol control and cardiovascular disease found from GWAS and the highest cause of death for NAFLD is heart disease, thereby pointing to a connection between the two diseases [136,137].

Another study by Liu et al. explored a gene-metabolite network analysis across non-alcoholic fatty liver, NASH and NAFLD with T2D in rat models [104]. Metabolomics and transcriptomics were carried out for rat liver and blood to identify DEGs and differential metabolites, followed by the construction of protein-protein and protein-compound interaction networks using STRING as the basic framework. With this method they found overlap between the three tested phenotypes for key regulators such as *CCL2* and *JUN*, which regulate genes mainly involved in inflammation and metabolism. Furthermore, the different NAFLD phenotypes also contained unique pathways, specifically lipid and fatty acid metabolism in the steatosis models, inflammatory and immune response in NASH models, and AMPK signaling pathway and insulin response in models involving T2D [104]. These results support the presence of both pathway perturbations fundamental to NAFLD across stages and pathways which may influence specific stages or conditions in NAFLD progression. 

### 5.3. Network Modeling of NAFLD Comorbidities

NAFLD is tightly associated with obesity, which is supported by various studies, with the supposed progression beginning with obesity which then develops into steatosis and subsequently NASH, and these developments are governed by multi-tissue changes (Figure 1). Gawrieh et al. carried out a global hepatic gene expression study on 53 morbidly obese individuals of which 27 had NAFLD and the remaining 26 acting as controls to identify DEGs [114]. To explain the connections amongst the DEGs in NAFLD, they implemented the Ingenuity Pathways Analysis to identify the interaction pathways between genes and their biological importance in NAFLD. In brief, significant DEGs were algorithmically linked based on an extensive IPA knowledge base into networks that were systematically ranked for relevance and analyzed to understand the biological function of the gene/disease. These DEGs showed importance in cellular movement, cell death, immunological disease, and lipid metabolism. Hub genes in the network included *COL1A1*, important in ECM restructuring (Table 1), and *IL10*, an important immune regulator also previously implicated in NAFLD by other studies [138]. 

Further investigation into the interconnection of obesity and NAFLD was carried out by Wang et al. in which their HPRD-based PPI networks revealed that compared to control, healthy-obese and steatotic subjects showed high degree of differential expression for the hub gene *PRK-CA*, which interacts with *EGFR* and *CDC42* [113]. Previous studies have implicated *EGFR* in stimulating the proliferation of stellate cells, which are heavily involved in the deposition of ECM in the liver. *CDC42* is involved in saturated fatty acid-induced c-JNK in hepatocytes, found in NASH. Overall, *PRK-CA, EGFR, CDC42,* and *VEGF-A* were found to be upregulated in this study and imply a role of focal adhesion in NAFLD development and obesity [113].

Another fundamental question regarding NAFLD is resolving the potential bidirectional relationship between NAFLD and MetS and identifying whether one is the consequence of the other or if both arise independently. Zhang et al. implemented a simplified BN model to measure the reciprocal causality of these diseases, in which epidemiological and health measurement data were imputed to create a network defining the relationships between the experimental variables (NAFLD, obesity, dyslipidemia etc.) [115]. With this unorthodox approach, they carried out the first bidirectional longitudinal study showcasing the reciprocal causality between MetS and NAFLD, with MetS having a greater effect on NAFLD than vice versa [115]. It was also indicated that obesity and dyslipidemia were key factors linking NAFLD and MetS, which corroborates many of the pieces discussed throughout this review.

Network approaches have also been explored to investigate the interconnectivity between NAFLD and Alzheimer’s Disease (AD), both being degenerative diseases that are significantly affected by lifestyle. In the study by Karbalaei et al., 332 and 1200 genes associated with NAFLD and AD respectively were extracted from the DisGeNet database [111]. The shared genes between diseases were then modeled using the STRING database to construct a PPI network. Among the forty-two enriched pathways in the network, carbohydrate metabolism, long fatty acid metabolism, and interleukin (IL) signaling pathways were assessed. Seven nodes which strongly linked gene modules (bottleneck hubs) were identified as possible therapeutic targets which include *IL6*, *AKT1*, *TP53*, *TNF*, *JUN*, *VEGFA*, and *PPARG*. Elevated levels of IL6 have been previously documented in patients with NAFLD [139] and AD [140]. Interestingly, increased expression of *TP53* has been implicated in AD, whereas p53 inhibition attenuates liver injury in NAFLD mouse models, which suggests an inverse relationship [111]. *JUN* is a documented therapeutic target of AD [141] and is shown to have increased expression in NAFLD, while also being identified as a therapeutic target in the Liu et al. network paper discussed above [104]. Additionally, Qi et al. also found the Amyloid Beta Precursor (APP) gene to be important in NAFLD, which is a key and well known gene in AD [107]. With both obesity and diabetes commonly intersecting AD and NAFLD separately, it is no surprise that AD and NAFLD share protein networks and pathways.

Despite studies investigating the connections between NAFLD and its commonalities such as T2D, obesity, MetS and AD, more comprehensive multi-tissue multi-omics modeling of network connections between diseases is still needed. As exemplified by Shu et al., novel pathways can be found and potential key regulators of multiple diseases can be pinpointed when multi-omics datasets across tens of tissues are integrated [88].

### 5.4. Network Modeling for Sex Differences in NAFLD

Network approaches have also been applied to examine the similarities and differences between sexes in the development of disease to help elucidate potential protection or exacerbation due to sex differences between gonadal males and females. To this end, Kurt et al. explored the sexual dimorphism present in NAFLD via a rigorous multi-tissue multi-omics study to find causal gene networks and key drivers (regulator genes) within these networks [116], using the Mergeomics pipeline [142,143]. Here, reconstruction of tissue specific co-expression modules/networks were built via complementary network methods—WGCNA [50] and MEGENA [51] utilizing gene expression data from liver and adipose tissues extracted from females and males in the HMDP cohort. The benefit of using both methods allows for one to cover for the other for potential missed biology and therefore captures more information to understand disease progression. Integrating these modules with sex specific GWAS data for liver steatosis as well as eQTLs for liver and adipose, they captured numerous pathways which were then mapped on Bayesian networks to predict key drivers of the disease. Among genetically perturbed pathways shared between the sexes, are immune system and metabolic pathways (branched chain amino acid metabolism/oxidative phosphorylation). For females, the perturbed sex-specific pathways were vitamin/cofactor metabolism and ion channel transport, while male pathways were related to phospholipid, lysophospholipid, and phosphatidylinositol metabolism and insulin signaling. Regulatory genes found to be sex-specific included *CHCHD6* in the male liver, which was also found as a key gene target in several previous studies (Table 1) [82,104,106,110] and was also recently validated to be important for the inhibition of glucose uptake and decreased mitochondrial activity in HepG2 cells [144]. For females they found tissue-specific genes such as *NCKAP1L*, an estrogen receptor target, and *TYROBP* (protein tyrosine kinase binding protein) as top key drivers for immune regulation in the female liver. Additionally, in female adipose tissue, the innate immunity linked genes *SH3BP2* and *C8B* were highlighted, thus in both tissues immune dysfunction was implicated in females [116,145].

Sexual dimorphism in NAFLD was also exampled in a recent study by Tomaš et al. in which they created a sex-based liver metabolism model called *LiverSex* derived from the *SteatoNet* liver computational model [146]. By modeling both cellular responses to sex hormones and the sex dependent variation in growth hormone release, the study helped identify key regulatory factors which confer the largest difference between sexes in hepatic triglyceride accumulation and are associated with the progression of NAFLD. Specifically, these include the more female specific *PGC_1_A* and *FXR*, and the more male specific *PPAR-alpha* and *LXR*. These studies made strides in the determination of sexual dimorphisms in pathologically relevant pathways and regulatory genes, thus pointing to potential avenues for sex specific NAFLD treatment in the future. 

### 5.5. Use of Network Models as Predictive and Diagnostic Tools for NAFLD development

In addition to improving mechanistic understanding of NAFLD, a network-based approach can also provide us with the opportunity to detect biomarkers, which can aid in a faster and non-invasive option for diagnosis of potential NAFLD patients. For detection of an unhealthy liver we have basic serum tests which look for abnormalities in markers to do with cell integrity ALT and AST, biliary tract function (Gamma GT and Alkaline Phosphatase) and functionality (Albumin). However, these detection methods, although indicating poor liver health, do not advise on the form of liver damage or severity of NAFLD progression accurately without a biopsy (Figure 1). Therefore, it is of key importance to find new biomarkers specific to NAFLD and its separate stages, which will assist with more accurate diagnosis and precision medicine. There have now been a few studies implementing differing network approaches to predict new biomarkers. 

To this end Mardinoglu et al. utilized GEM modeling of hepatocytes via the incorporation of the HMR2 database, proteomics, literature and clinical data to identify biomarkers in NASH [98]. Here, they found blood concentrations of heparin sulfates, chondroitin and serine deficiency of potential diagnostic use. Moreover, they provide insight into druggable targets for NASH such as the metabolism related *BCAT1* and *SHMT1* genes and the serine linked *PSPH* gene [65]. Zhu et al. implemented another variant of network modeling for new biomarkers of NAFLD via transcriptomics of rat liver by constructing miRNA-mRNA networks as well as constructing PPI (STRING) and pathway interaction networks, finding hub genes playing key roles in metabolism such as *Cyp1a1*, *Cyp51,* and *Hmgcr*, which have been previously correlated with NAFLD [98]. 

Network models can additionally be integrated to predict how certain environmental contributions such as diet may affect disease progression. Maldonado et al. used a combination of quantitative kinetic regulatory networks with qualitative GRNs via the quasi-steady state Petri nets (QSSPN) method [147] to investigate whether glucose or fructose was worse for NAFLD progression [97]. They found that there was no difference in lipogenic consequences between fructose and glucose and increasing either sugar enhanced lipid generation. Additionally, they aimed to predict the impact of activating PPARα by lipids and showed that PPARα activation appears to have a significant role in hepatic steatosis progression, suggesting issues with PPARα agonists as treatment options [97].

Overall, these different approaches showcase the flexibility of network modeling to be applied to NAFLD predictions. Notably, the use of different types of data and network approaches yielded different biomarkers, thereby calling for comprehensive comparison and validation efforts.

### 5.6. Network Insights into Drug Repositioning for NAFLD Treatment

Drug repositioning has proven useful for numerous diseases and computational packages have been developed to predict the possibility of repositioning drugs with known safety profiles hence providing a better chance for translational success. With no current pharmacological options for NAFLD but a large association with other diseases such as obesity and T2D, it begs the question that perhaps treatments for such diseases may provide significant therapeutic benefits for NAFLD. Although direct predictions for drug repositioning based on NAFLD molecular networks are yet to be carried out, various recent studies have used network modeling to understand how drugs already tested may provide a therapeutic benefit for NAFLD [148].

Barbosa et al. investigated the potential benefits of the T2D drug liraglutide (GLP-1 receptor agonist) on hepatic steatosis treatment [118]. Specifically, the group assayed levels of key blood parameters such as cholesterol and triglycerides, carried out a transcriptomic analysis to provide potential target genes for liraglutide and liver steatosis to assess interaction networks. By treating NAFLD model mice with liraglutide and comparing the gene network between these and control, the study identified *S6K1* as being central to steatosis. Moreover, the study showed therapeutic promise with the drug bettering key metabolic parameters and reducing the content of liver fat compared to controls [118].

Singh et al. explored a T2D drug, thiazolidinedione (TZD) and other drug treatments (cardiovascular treatment drug pentoxifylline, obeticholic acid, and vitamin E with placebo) [119]. They implemented a Bayesian network meta-analysis to investigate the effectiveness of these pharmacological agents in the treatment of NASH patients via direct (comparing between treatments) and indirect (comparing treatments of interest with a common comparator, such as placebo) comparisons. Their findings implicate that these therapeutics had positive results in reducing fibrosis and ballooning degeneration in NASH patients [119].

The effectiveness of pure total flavonoids from citrus (PTFC) particularly the components naringin, neohesperidin and narirutin, in the treatment of NASH has been recently tested via a combination of WGCNA, STRING, and the CTD (comparative toxicogenomics database) analyses [117]. The study revealed one of the underlying core genes targeted by PTFC in NAFLD progression through the analysis of high fat diet mice groups with or without PTFC treatment. Based on the common nodes between networks, VEGF-C, a component of cytokine-cytokine receptor interaction and focal adhesion pathways, was implicated to play an important role in disease progression and is a target of PTFC.

Despite showing some promise, these specific therapeutics show low efficacy overall with limited progress in clinical testing. They are also based on prior hypotheses that certain drugs may be useful for NAFLD rather than based on data-driven approaches that may shed light on novel drug predictions, thus highlighting the further need for integrated systems biology models [149]. Particularly, by better elucidating the species- and tissue-specific network models of NAFLD and leveraging species- and tissue-specific drug signatures, computational models may be able to create networks that more accurately link causal disease networks to drug targets, thereby making better predictions concerning drug repositioning for NAFLD.

### 5.7. NAFLD Insights Learned From Network Modeling so Far

When viewing the outcomes of the various network modeling methods in a generalized manner one can begin to isolate some key mechanisms which drive NAFLD (Figure 3). In particular, many of the GRN-based studies identify pathways closely associated with ECM remodeling, immune activation, and mitochondrial function, particularly within the context of lipid metabolism. Furthermore, GEM based models have identified specific metabolites such as cholesterol, glutathione and NAD+ as dysregulated in NAFLD, indicating the pathogenic role of specific metabolic loci. The major pathways identified consistently in single tissue network studies, are also indicated in multi-tissue studies, particularly metabolism and inflammation, which are likely to be systematically dysregulated in NAFLD. 

Numerous regulatory genes have been identified which fall under the scope of metabolic regulators, including *NR5A2*, a key regulator in cholesterol homeostasis, and *PLIN2*, *CIDEC*, and *DNAAF1* which suppressed expression of *PKLR* and *CPT1A*, genes involved in mitochondrial metabolism and functionality. Additionally, *in vivo* studies further implicate dysregulation of *PKLR* in mitochondrial dysfunction. Several groups also indicate genes directly involved in lipid metabolism, immune function, and cell cycle such as *PCSK9*, *PNPLA3, CCL2*, and *JUN*, as mechanistic drivers of NAFLD. Moreover, metabolic and immune pathways are also heavily implicated in the sexual dimorphism of NAFLD, as males are indicated to associate more strongly than females with pathways related to phospholipid, lysophospholipid, phosphatidylinositol metabolism, and insulin signaling, and females and males demonstrated differential tissue specificity for various immune pathways.

Continuing the discussion of sex differences in NAFLD, network studies, particularly the study by Kurt et al. have identified several sexually dimorphic pathways in liver and adipose tissue of mice. Specifically, alterations in vitamin/cofactor metabolism and ion channel transport in females. This study and others also identify key genomic loci which show sex biased expressional patterns in relevant tissues, thus providing further suggestive loci for future study. In a broader attempt to categorize these changes, *LiverSex* is a promising tool which enables users to computationally identify sex dependent expressional variations in disease, thus providing further resources for research along this path. 

Network modeling has not only furthered the mechanistic understanding of NAFLD, but also provided insights into the disease specific biomarkers which may be utilized to assist healthcare professionals in earlier screening and more specific predictions regarding disease progression such as analysis of heparin sulfates, chondroitin and serine deficiencies. 

Overall, network studies have revealed the importance of numerous tissue-specific and cross-tissue pathways, regulatory molecules, and predictive markers. However, the coverage for tissues and omics domains remains limited and few of the findings have led to translation in ameliorating the incidence or progression of NAFLD, thus there is ample room for discovery. Additionally, many of the network studies of NAFLD show a lack of consistency in terms of species used, stage of disease examined, omics data types included, and network approach used, which adds to the challenge of finding definite trends across studies. 

## 6. Future Directions for Network Modeling in NAFLD

As highlighted above, many of the current methods utilize a diverse set of computational tools and different omics datasets to elucidate the genes and pathways contributing to NAFLD. While this has a potential strength of increasing the odds of capturing diverse causal mechanisms, this disjointed approach hinders straightforward comparison between results. Thus, there is a need for comprehensive tissue and omics data collection across studies and systematic method comparison to derive unification algorithms which can be applied across studies to facilitate reproducible and comparable research.

Given that both strong genetic and environmental factors are major contributors to NAFLD, understanding the relative contributions of these risks, the divergence and similarities in the networks and pathways perturbed by these risks, and establishing network connections between these two major components will be a key future area of focus. Furthermore, as NAFLD disease progression is classified into distinct stages, future studies are needed to identify either the distinct network perturbations under each stage which drives the sequential evolution of the disease or highlight fundamental molecular changes which are a common denominator across all stages of this disease. Such predictions will offer essential insight into how NAFLD progresses and what components of that progression encompass the genetic, epigenetic, or environmental factors influencing pathogenesis. 

As NAFLD has been examined in both human populations and animal models, a systematic comparison between species is also needed to assess the translational potential of animal model studies. Such comparison will require similar study designs targeting genetic or environmental factors, inclusion of similar sets of tissues and multi-omics datasets, and similar network modeling approaches to enable tissue-specific comparisons between species at the gene network level.

Finally, significant progress can be made within the scope of multi-tissue interplay, single cell analysis, and drug repositioning. To date we only identify a few studies that investigate tissues outside of the liver, which currently neglects the interplay and dynamics of other tissues contributing to NAFLD progression. Moreover, single cell multi-omics is the next frontier in this area due to the heterogeneous nature of tissues as well as disease progression. Single cell analyses can reveal cellular mechanisms at a much higher resolution than at the bulk tissue level but to date only one single cell study has been conducted in this area. As the specificity of data regarding disease mechanism grows, a more systematic drug repositioning effort leveraging existing omics information of approved drugs with known safety profile and molecular pathways of NAFLD is required. Success within this realm will provide the potential to supply treatment options for NAFLD at a much faster pace.

## 7. Concluding Remarks

NAFLD is a common complex disorder that engulfs many stages of disease. It is commonly associated with other metabolic syndromes and multiple tissues, which implies the holistic setting that is required for disease progression. With a multifactorial disease, we need to examine the multiple contributions leading to the pathogenic endpoint. A multi-tissue multi-omics systems biology approach can provide us with information of the many facades of the disease and we can examine these omics sets using network modeling algorithms to predict biomarkers, hub genes, pathogenic pathways and potential therapeutic targets. The benefits of this systems approach are overwhelming as it offers a more comprehensive understanding of the pathophysiology accounting for both internal and external contributors at multiple scales, which is critical for guiding novel therapeutics for NAFLD, an alarmingly growing health concern.

## Figures and Tables

**Figure 1 genes-10-00966-f001:**
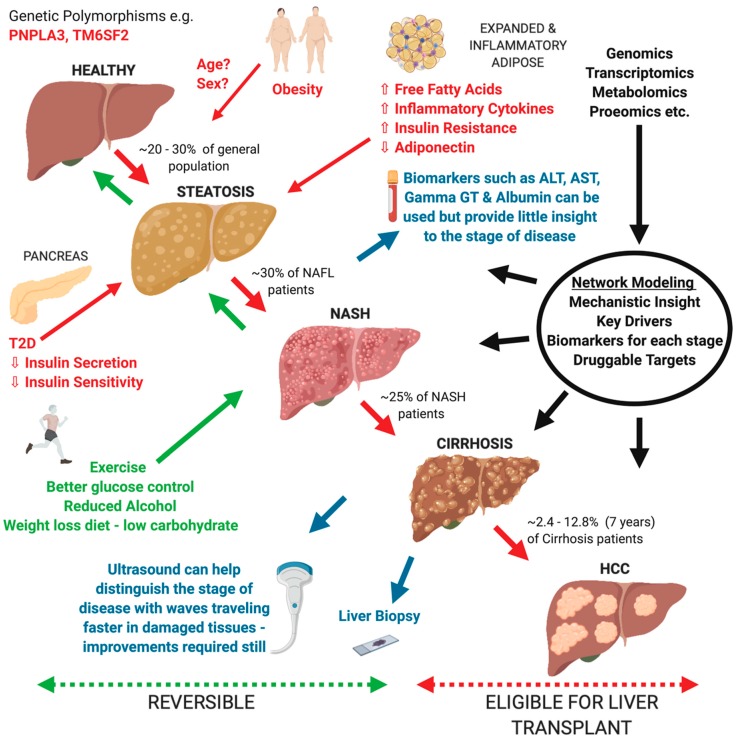
Overview of our current understanding of Non-alcoholic fatty liver disease (NAFLD) progression from a healthy liver to hepatocellular carcinoma (HCC). The red items indicate the different potential contributors towards NAFLD as well the progression from a healthy liver to HCC. The green items indicate potential disease reversibility from non-alcoholic steatohepatitis (NASH) to a healthy liver, through different methods such as exercise and better glucose control. The blue items showcase the various tests that can be utilized to investigate liver health. The black items represent the omics approaches and use of network modeling to address various knowledge gaps for NAFLD.

**Figure 2 genes-10-00966-f002:**
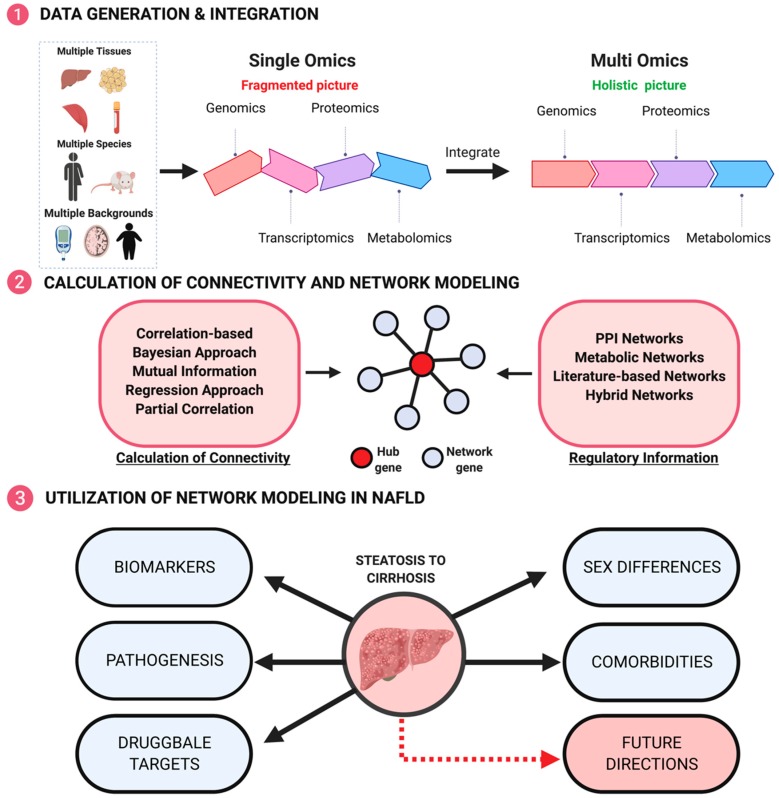
Overview of the methodological workflow in applying network approaches to study NAFLD. Step 1. Through collecting various tissues from multiple species across multiple disease backgrounds, we can run different omics analysis to provide a useful but one-dimension view of what is occurring in the disease state. However, by combining multiple omics datasets, we can provide a holistic picture. Step 2. We can build networks through various calculations of connectivity and regulatory information to elucidate hub genes in disease networks. Step 3. Once networks are built, we can utilize them to address the gaps in disease understanding and therapeutic discovery.

**Figure 3 genes-10-00966-f003:**
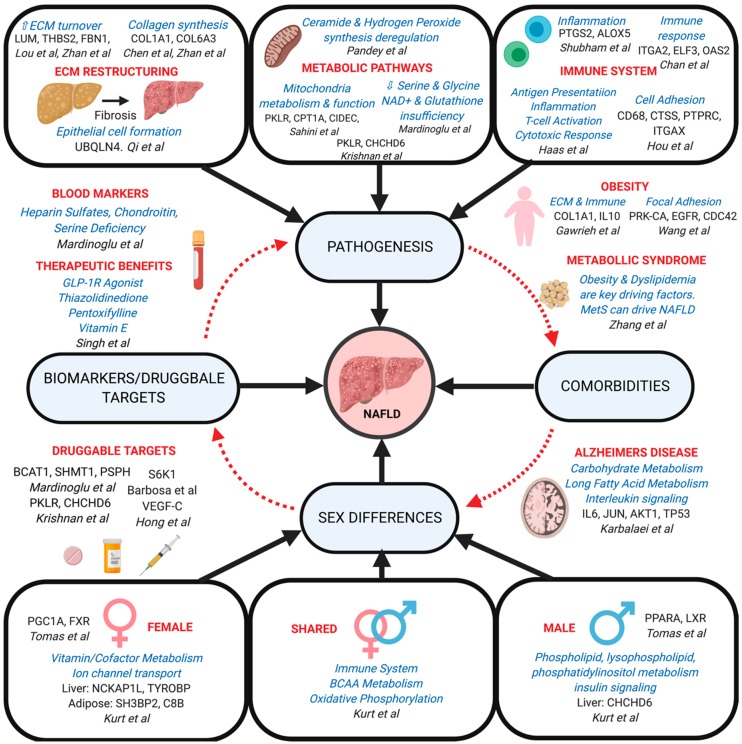
Summary of findings from network-based studies to elucidate NAFLD progression, mechanisms, comorbidities, sex differences, biomarkers and druggable targets.

**Table 1 genes-10-00966-t001:** Application examples of network approaches to NAFLD. Shared pathways and genes across studies are highlighted in bold.

Application categories	Paper	Network Method	Omics Data	Tissue	Pathways	Key Drivers/ Findings
**Prediction**	Maldonado et al. [97]	ODE, GEM, and QSSPN Stochastic Simulation	Transcriptomics and Proteomics	Liver (human)	**Lipid metabolism**	No significant difference between glucose and fructose on lipogenesis. PPARα activation implicated in steatosis
	Zhu et al. [98]	STRING and miRWalk	Transcriptomics	Liver (rat)	**Lipid metabolism** (steroid synthesis, fat digestion, bile secretion) **Inflammatory/Immune response** (NF-κB, cytokine signaling)	Abcg8, Cyp1a1, Cyp51, **Hmgcr**, Acat2, Cyp7a1, **Cyp7b1**, **Cd36**, Cd44, RT1-Da, RT1-Ba, RT1-Bb, RT1-Db1
	Ma et al. [99]	Bayesian	N/A	N/A	N/A	Bayesian network provides screening and predictive value for NAFLD
**Pathogenesis**	Pandey et al. [72]	MiNEA and GEM	Metabolomics, Proteomics and Transcriptomics	Liver (mouse and human)	**Inflammatory/Immune response** (ceramide synthesis, oxidative stress) **Lipid metabolism** (cholesterol synthesis)	Identified deregulation in metabolic networks of ceramide and hydrogen peroxide synthesis for NASH in both mice and humans
	Mardinoglu et al. [65]	GEM	Metabolomics and Proteomics	Liver (human)	**Lipid metabolism** (peroxisome, steroid biosynthesis, FA biosynthesis) **Protein metabolism** (amino acid turnover) **Inflammatory/Immune response Mitochondrial stress**	Constructed a consensus GEM for hepatocytes termed iHepatocytes2322, which was used to identify serine deficiency in NASH and possible therapeutic targets PSPH, SHMT1 and BCAT1
	Shubham et al. [100]	WGCNA and GEM	Transcriptomics	Visceral Adipose Tissue (human)	**Lipid metabolismInflammatory/Immune response** (hypoxia) **ECM remodeling** **Protein metabolism**	FOSL1, HIF1A, CHSY1, NAMPT, NAMP, NCOR2, SUV39H1, SUV420H1, CHD9, CAT, ALDH2, HADH, ETFA, ETFB, PPRC1, CYP2C8, ADH4, DAPK1
	Mardinoglu et al. [101]	GEM	Metabolomics and Proteomics	Liver and Adipose (human)	**Lipid metabolism** **Carbohydrate metabolism** GSH biosynthesis NAD+ repletion	Dietary supplementation of GSH and NAD+ precursors are possible NAFLD treatment options
	Hou et al. [102]	STRING	Transcriptomics	Liver (mouse)	**Lipid metabolism** (steroid synthesis, FA activation) **Cell cycle** (PI3K–Akt signaling) **Inflammatory/Immune response** (phagocytosis)	Itgb2, Hck, Rac2, CD48 Ptprc, **Jun**, Cd68, **Tyrobp**, Ctss, Itgax, Hsd3b5, Cyp2c44 Cyp2c54, Cyp1a2, Cyp2c70, Ugt2b1, **C8b**, **Egfr**, Gyp7b1, Slco1a1
	Lou et al. [103]	WGCNA	Transcriptomics	Liver (human)	**Lipid metabolism** **Inflammatory/Immune response** (fibrosis) **Cell cycle** (PI3K-Akt signaling) **Cell adhesion** **ECM remodeling**	**LUM**, **THBS2**, **FBN1**, EFEMP1, SELENBP1
	Liu et al. [104]	STRING	Transcriptomics and Metabolomics	Liver and Blood (rat)	**Lipid metabolism** **Inflammatory/Immune response** (TNF, cytokine signaling, TLR signaling) **Cell cycle** (NF-κB, p53 signaling) **ECM remodeling** **Mitochondrial stress**	**Jun**, Ccl2, Ccl12, Icam1, Cxcl2, Cdkn1a, Serpine1, Rprm, **Fabp4**, Fabp5, Bcl2a1, **Cxcl10**, Olr1
	Krishnan et al. [82]	WGCNA, MEGENA and Bayesian	Genomics and Transcriptomics	Liver and Adipose (mouse)	**Lipid metabolism** **Cell cycle Inflammatory/Immune response** (peroxisomal pathways) **Insulin signaling** **Carbohydrate metabolism** (TCA cycle) **Apoptosis**	*Fasn, Pklr, Chchd6, Thrsp, Cd36, Acly, Hmgcr, Acaca, Acacb, Col1a2, Elovl6, Ptpn6, Echs1*
	Xiong et al. (Single cell) [105]	Ligand-Receptor Interaction (Fantom 5)	Transcriptomics and Proteomics	Liver (mouse)	**Lipid metabolism** **Inflammatory/Immune response** (cytokine signaling) **Cell adhesion** **ECM remodeling**	Hepatic stellate cells serve as a hub of intrahepatic cell signaling. Identification of novel Trem2+ NASH-associated macrophages (NAMs) linked to disease severity
	Sahini et al. [106]	STRING	Genomics and Transcriptomics	Liver (human)	**Lipid metabolism** (lipogenesis, lipid droplet growth) **Inflammatory/Immune response** **Carbohydrate metabolism** **Mitochondrial stress**	PLIN2, **CIDEC**, HILPDA, STAT1
	Qi et al. [107]	PPI (HPRD)	Transcriptomics	Liver (human)	**Inflammatory/Immune response** (cytokine activity) **Cell adhesion**	*UBQLN4, APP, SHBG, CTNNB1, COL1A1*
	Chan et al. [108]	MetaCore	Transcriptomics	Liver (human)	**Lipid metabolism Carbohydrate metabolism** **Inflammatory/Immune response** **Cell cycle ECM remodeling** (fibrosis)	Identified 87 “significant” genes (not hub genes) associated with cirrhosis
	Chen et al. [109]	miRWalk and STRING	Transcriptomics	Liver (human)	**ECM remodeling** (focal adhesion, fibrosis) **Cell cycle** (PI3K-Akt signaling)	*COL6A1, COL6A2, COL6A3, PIK3R3, COL1A1, CCND2*
	Lee et al. [110]	Co-expression, PPI, TR and GEM	Metabolomics, Proteomics and Transcriptomics	Liver, Adipose, and Muscle (human)	**Lipid metabolism** (FASN enzyme activity) **Inflammatory/Immune response** **Mitochondrial stress** (transport) **Cell Cycle**	**FASN**, **PKLR**, **PNPLA3**, PCSK9
**Comorbidities**	Karbalaei et al. (AD) [111]	STRING (DisGeNet)	Genomics, Transcriptomics and Proteomics	Not specified (human)	**Lipid metabolism** **Carbohydrate metabolism** **Insulin signaling** Regulation of JAK-STAT **Inflammatory/Immune response** (IL signaling)	IL6, **AKT1**, TP53, TNF **JUN**, **VEGFA**, PPARG, MAPK3, IGF1, LEP
	Haas et al. (Obesity) [112]	WGCNA	Transcriptomics	Liver (human)	**Inflammatory/Immune response**	*CXCL9, CXCL10,* and *LYZ* higher expression in patients with NASH than steatosis
	Wang et al. (Obesity) [113]	PPI (HPRD)	Transcriptomics	Liver (human)	**Cell cycle** **Cell adhesion** **Protein metabolism** **Inflammatory/Immune response** (phagocytosis)	*PRKCA, EGFR, CDC42, VEGFA, CRK*
	Gawrieh et al. (Obesity) [114]	Ingenuity Pathway Analysis (IPA)	Transcriptomics	Liver (human)	**Lipid metabolism** **Cell cycle** (development, movement) **Apoptosis** (ubiquitination) **Inflammatory/Immune response** (cell death)	**COL1A1**, IL10, **DCN**, IGFBP3, HSPA5, USP25, **FABP4**, PPFIBP1, ZAK, RGN, SMUG1, CYP4F22, CSN2
	Zhang et al. (MetS) [115]	Bayesian	N/A	N/A	**Inflammatory response** (oxidative stress) **Insulin signaling** **Lipid metabolism** (dyslipidemia)	The effect of MetS on NAFLD is significantly greater than that of NAFLD on MetS
**Sex Differences**	Kurt et al. [116]	WGCNA, MEGENA and Bayesian	Genomics and Transcriptomics	Liver and Adipose (mouse and human)	**Lipid metabolism** **Protein metabolism** **Insulin signaling** **Inflammatory/Immune response** **Cell cycle** **Apoptosis** ECM remodeling	*AHSG, FASN, RBP4, SREBF1, ACOT2, DECR1, DHCR7, SQLE, INSIG1, ACSS2, BCKDHA, MCCC1, ECHS1, DCA8, MKI67, CCNA2, FBN1, COL1A2, CCDC80, RELB, IFNG, CXCL10, PTPRO, SH3BP2, TYROBP, C8B, CD36, CPT2, CHCHD6, GYS1, INPP5D, FCER1G, NCKAP1L, ANXA2, CIDEC, PNPLA3, TRIB1, CY7B1, HCK, FGL2, SLC2A3*
**Drugs**	Hong et al. (PTFC) [117]	WGCNA and STRING	Transcriptomics and Toxicogenomics	Liver (mouse)	**Lipid metabolism** **Cell cycle** (activation) **Cell adhesion Inflammatory/Immune response** (Toll-like receptor, cytokine and chemokine signaling) **Apoptosis**	VEGF-C and **COL4A1** may play a regulatory role in NAFLD development and are possible targets of PTFC
	Barbosa et al. (GLP-1 Receptor Agonist - Liraglutide) [118]	STRING and STITCH	Transcriptomics and Proteomics	Liver (mouse)	**Insulin signaling** **Cell cycle** **Inflammatory/Immune response** **Lipid metabolism**	*AKT1*, *RPS6KB1/S6K1*; Liraglutide decreases liver fat content and improves metabolic conditions
	Singh et al. (Vitamin E, Pentoxifylline, Obeticholic Acid and TZDs) [119]	Bayesian	N/A	N/A	N/A	Pentoxifylline and Obeticholic Acid improve fibrosis. Vitamin E, TZDs, and Obeticholic Acid improve ballooning degeneration in NASH patients

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
