# Peer review of "Network Modeling Approaches and Applications to Unravelling Non-Alcoholic Fatty Liver Disease"

_genes, 2019, doi:10.3390/genes10120966_

Round 1
Reviewer 1 Report
In this manuscript, the authors put together a comprehensive, clear and well-structured review on Non-Alcoholic Fatty Liver Disease (NAFLD). I found the table and the figures very informative. I believe this review would be well received by the broad readership of “Genes” and I recommend minor revisions.
Remarks:
Define what network with scale-free topology are (line 299). Perhaps expanding on the description of networks (given that it is so central to the review), common metrics used to describe them, etc may be beneficial particularly for readers less familiar with the computational aspects of network analysis. The authors introduced several network-based approaches in section “Commonly Used Network Models”. I wonder if the authors could elaborate further what stated at lines 411-413 and provide some concrete examples of how some of these network approaches could be used together and what specific insights could be gained by each method. I believe providing this kind of clarifications/guidelines would guide readers in selecting the method(s) that best suits their needs and would increase the impact of this review.
Minor:
Typo in “Tanscriptomics” (line 255) Typo in “repoisioning” (line 447) Figure 2 is referenced before Figure 1 in the text
Author Response
We thank the reviewer for the positive comments and constructive suggestions provided. We have thoroughly addressed each comment in the revision. Below are our point-by-point responses, with the reviewer comments in black and our responses in blue.
Define what network with scale-free topology are (line 299). Perhaps expanding on the description of networks (given that it is so central to the review), common metrics used to describe them, etc may be beneficial particularly for readers less familiar with the computational aspects of network analysis.
Response: We appreciate the comments and have provided a clearer definition of scale-free topology as suggested and have expanded the descriptions of the networks in the revised manuscript.
The authors introduced several network-based approaches in section “Commonly Used Network Models”. I wonder if the authors could elaborate further what stated at lines 411-413 and provide some concrete examples of how some of these network approaches could be used together and what specific insights could be gained by each method
Response: As suggested, we have elaborated the network approaches with more details and provided some general guidance and concrete examples of how some of the network approaches could be used together to provide novel insights in disease pathogenesis.
Typo in “Tanscriptomics” (line 255) Typo in “repoisioning” (line 447) Figure 2 is referenced before Figure 1 in the text
Response: Thank you for pointing out these typos. We have corrected the above typos and thoroughly revised the manuscript to improve both grammar and spelling. We also added the citation of Figure 1 before Figure 2.
Reviewer 2 Report
This review analyzes an important number of works dedicated to Non-alcoholic fatty liver disease (NAFLD) highlighting the many results obtained with different techniques and underlining some structural limits such as the non-homogeneity of the biological systems used (different animal models, conditions pathological not always homogeneous, etc).
It is extremely interesting to see the developments obtained with very different applications of computational methods according to the complex etiology of the disease.
Authors provide general overviews by well organized figures and table.
Small revisions are needed in the text to eliminate the various typos and to better arrange the references: in some cases they seem to be missing (eg lines 104-105) and in some others they do not seem to be in the best place (eg line 182 for TM6SF2).
Figure legends should be improved: for instance in Figure 1, the use of different colors should be explained in order to make the understanding of images even more immediate.
Maybe authors could add a new table where genes or proteins found as drivers or important players for the disease are the starting points and the information about the type of biological context and the type of model (both animal and mathematical) used is reported: in this case the possibility to see if a same gene (and the related pathways) was found in different studies would be easier.
Author Response
We appreciate the reviewer’s positive comments and constructive suggestions. We have taken them into account for this revision. Below are our point-by-point responses, with the reviewer comments in black and our responses in blue.
Small revisions are needed in the text to eliminate the various typos and to better arrange the references: in some cases they seem to be missing (eg lines 104-105) and in some others they do not seem to be in the best place (eg line 182 for TM6SF2).
Response: Thank you for your comments. We have corrected the various typos and the placement of the references throughout the manuscript. We have also added in the missing references in the lines 104-105.
Figure legends should be improved: for instance in Figure 1, the use of different colors should be explained in order to make the understanding of images even more immediate.
Response: We appreciate the comments and have elaborated the figure legends, particularly we have defined the use of colors within Figure 1 as the reviewer suggested and explained the three steps in Figure 2 to provide further context to the figure.
Maybe authors could add a new table where genes or proteins found as drivers or important players for the disease are the starting points and the information about the type of biological context and the type of model (both animal and mathematical) used is reported: in this case the possibility to see if a same gene (and the related pathways) was found in different studies would be easier.
Response: We agree with the reviewer that it would be better to showcase the same genes occurring across different studies. Therefore, to improve the ability of the reader to find consistent genes/drivers within the network studies we have made bold any gene/driver within the table that had appeared in at least two studies. We additionally made bold the consistent pathways to further highlight the key pathways involved in NAFLD.